# Functionalized Carbon Nanoparticles as Theranostic Agents and Their Future Clinical Utility in Oncology

**DOI:** 10.3390/bioengineering10010108

**Published:** 2023-01-12

**Authors:** Seung S. Lee, Miltiadis Paliouras, Mark A. Trifiro

**Affiliations:** 1Division of Experimental Medicine, McGill University, Montreal, QC H4A 3J1, Canada; 2Lady Davis Institute for Medical Research—Jewish General Hospital, Montreal, QC H4A 3J1, Canada; 3Department of Medicine, McGill University, Montreal, QC H4A 3J1, Canada; 4Department of Oncology, McGill University, Montreal, QC H4A 3J1, Canada

**Keywords:** carbon nanoparticles, theranostics, physical-directed therapies, tumor ablation, imaging, cancer, genetic heterogeneity, clinical trials

## Abstract

Over the years, research of nanoparticle applications in pre-clinical and clinical applications has greatly advanced our therapeutic and imaging approaches to many diseases, most notably neoplastic disorders. In particular, the innate properties of inorganic nanomaterials, such as gold and iron oxide, as well as carbon-based nanoparticles, have provided the greatest opportunities in cancer theranostics. Carbon nanoparticles can be used as carriers of biological agents to enhance the therapeutic index at a tumor site. Alternatively, they can also be combined with external stimuli, such as light, to induce irreversible physical damaging effects on cells. In this review, the recent advances in carbon nanoparticles and their use in cancer theranostics will be discussed. In addition, the set of evaluations that will be required during their transition from laboratory investigations toward clinical trials will be addressed.

## 1. Introduction

The use of nanomaterials in cancer theranostics has become of great interest over the last few decades. Because many nanoparticles are in the range of 10–100 nm in size, their half-lives in circulation are greatly enhanced compared to small active pharmaceutical drugs [1]. In addition, these nano-formulations provide additional benefits, such as alleviating off-target toxicity and evading the need to use strong excipients to stabilize emulsions of nonpolar drugs in an aqueous environment [2,3]. The earliest forms of nanoparticles in nanomedicine were focused on drug delivery to enhance the therapeutic index of pharmaceutical agents. For example, formulations, such as liposomal doxorubicin (Doxil) and albumin-bound paclitaxel (Abraxane) are two of the well-known clinically approved nanoparticles that have been used against different types of cancers with success [4,5]. However, the greatest challenge to cancer therapeutics has become genetic heterogeneity which allows neoplasia to genetically evolve and become resistant to treatment regimes [6,7,8,9]. Although significant advances have been made in cancer biological therapeutics, fundamentally, they have not contributed to the development of any alternatives toward addressing genetic heterogeneity.

In recent years, the utilization of nanoparticles has extended beyond the classical drug delivery methods by exploiting their innate properties. Such unique properties of nanomaterials include but are not limited to, utilizing their high surface area for the improved cargo loading, enhanced magnetic properties, and unique optical characteristics, which may be further manipulated by external stimuli, such as pH, temperature, light, and magnetic fields [10,11]. There are several applications using nanoparticles as a component to deliver physical-directed therapies. Several research groups have used nanoparticles for photothermal therapy by direct tumor injection [12,13,14,15,16]. While the technical difficulties of direct intratumoral injection could be a challenge, the approach would allow for a significantly higher concentration of nanoparticles localized in the tumor site [17]. However, a consequence of intratumoral injection combined with photothermal therapy is the generation of excessive bulk heating, which leads to extended and unwanted damage to normal tissue that would result in serious injury [12,13,14,15,16,18,19,20]. Carbon nanoparticles (CNPs), like other inorganic nanoparticles, absorb light with extreme efficiency generating high surface temperatures. If placed adjacent to cancer cells, extensive hyperthermic ablation is achieved [21,22,23]. They can also be uniquely functionalized on their surface. For example, a multi-walled carbon nanotube with a diameter of 50 nm and a length of 1.5 µm will have approximately 10^4^ attachment sites per single particle, and that can be occupied with targeting and/or imaging moiety. By using these targeting measures, a critical mass of nanoparticles can be accrued at a tumor site and, in conjunction with light irradiation, can confer very focal tumor ablation. CNPs absorb light with extreme efficiency and generate high surface temperatures (400 °C) in femtoseconds (fs) [24,25]. If targeted at cancer cells, ablation will be sufficient to damage the targeted tumor cells but also the neighboring cells resulting in a “near-field” effect during treatment. The significance of this near-field effect on tumor-adjacent cells cannot be understated when acknowledging the transformative influence cancerous cells have through the tumor microenvironment, and the microenvironment itself is a risk factor for new neoplasia in situ [26,27]. Another acknowledgment of phototherapy is the activation of immune responses, as a result of releasing intracellular damage-associated molecular patterns (DAMPs) from the tumors and increasing the immunogenicity of the tumor microenvironment [28,29,30]. The release of DAMPs induces the maturation of dendritic cells, which will migrate to the lymph nodes where the cross-penetration of antigen to differentiate naïve T cells into CD8+ T cells [31]. In addition, certain cytokines (IFNγ, TNFα, and various interleukins) have been reported to increase. The activation of the immune response would have a positive effect on removing the primary tumor and residual tumor cells [32].

Ultimately, research performed using a variety of different nanomaterials is aimed at meeting the recognized shortcomings of cancer therapeutics. In this review, we will describe the applications of CNPs as theranostic tools in cancer and discuss the challenges faced by all nanomaterials as they advance toward clinical trials.

## 2. General Overview—Nanoparticles and Theranostic Applications

*(i) Nanoparticles as carriers of therapeutic payload.* In addition to the classical liposomes and micelles, different nanomaterials have been used to load molecules of interest. Using different methods, payloads have been loaded onto nanoparticles for enhanced delivery to malignant tissues. Physical, non-covalent interactions, such as van der Waals forces, hydrogen bonding, hydrophobic interactions, pi stacking, and different electrostatic interactions, have all been utilized to load drugs onto different nanomaterials [33]. Not only such preparations are reversible, but such molecules are also not chemically modified; thus, they can remain active in their native forms. However, as a result of the non-covalent interaction, the drug compound may become detached and reduce the delivery of the effective concentration at the tumor site or elicit distant off-target or adverse side effects. On the other hand, irreversible covalent conjugation of molecules of interest or utilizing stronger interactions, such as avidin–biotin complexes, are preferred in other situations [34]. Covalent conjugation often requires the use of coupling agents and crosslinkers to form bonds between functional groups, such as carboxylic acids, amines, thiols, maleimides, hydrazines, aldehydes, and more [35]. These nanoparticle-bound molecules would maximize the therapeutic index of the pharmaceutical agent. Alternatively, the nanoparticle-payload conjugate may be internalized to the cells via endocytosis, eliciting physiological damage to cancer cells [36,37].

*(ii) Nanoparticles responding to external stimuli.* Whether employing a drug payload delivery platform or exploiting a physical attribute of a nanoparticle as a cancer therapeutic modality depends on several factors, including accumulating and retaining the nanoparticle at the tumor site. These can be accomplished by specifically formulating the nanoparticles, such as optimizing the size of the nanoparticle or surface modification to enhance their half-life (i.e., PEGylation or albumin) and/or including a tumor-specific targeting moiety. Another means would be also exploiting the pathophysiological characteristics of the tumor, such as enhanced permeability and retention (EPR). Together this would combine maximizing the concentration with the activating property of the nanoparticle at the tumor site (Figure 1). Activation methods that are being currently employed in preclinical and clinical settings to induce or enhance treatment efficacy include exploiting the pH differences between normal tissue and the tumor microenvironment or amplifying the potential therapeutic effects through external means by applying light, magnetic fields, and ultrasound.

Acidic Microenvironment. The abnormal metabolism and protein regulation in the tumor microenvironment favors anaerobic glycolysis, thereby making the region more acidic than normal tissues, with the microenvironment of the tumor typically ranging from pH 5.5 to 6.5 [38,39,40,41,42]. In addition, the pH of lysosomes, where different biomolecules are broken down, is in the range between 4.5~5 for normal cells and 3.5~5 for cancer cells [43]. Accordingly, nanoparticles that selectively release therapeutic agents under acidic conditions have been designed. Nevertheless, the kinetics of pH-based drug adsorption, though it is often influenced by many physiological factors, some investigators have reported that the total accumulation of drugs at the tumor site was not significantly different between acid-cleavable nanocarriers and noncleavable nanocarriers [44].

Light. The application of light in biomedicine has been essential for confocal microscopy and imaging biopsies [45], and more recently in near-infrared spectroscopy, such as neuroimaging the extent of cerebral oxygenation and hemodynamics [46]. Light may be used to induce damage to tissues via phototherapies. This can be achieved with light either creating radical oxygens and thereby provoking cellular cytotoxicity or by thermally ablating the cells of the exposed region. Often laser light is guided to deliver a sufficient amount of energy at a particular location leading to enucleation or vaporization of the exposed tissues [47]. These laser-guided tissue ablation techniques can be further enhanced by using molecules that exhibit strong optical properties. Phototherapy for cancers was first introduced in the 1970s as a photodynamic therapy [48,49]. Photodynamic therapy uses photosensitizers (e.g., Porfimer sodium, verteporfin, and temoporfin) introduced to the body (usually not targeted), followed by light therapy, whereupon the photosensitizers’ absorbance of light (usually visible light wavelength) leads to chemical production of cytotoxic radicals, such as reactive oxygen species that cause cell apoptosis [50,51]. However, photodynamic therapy is very limited in scope and cannot lead to significant tumor ablation [52,53]. Furthermore, most photodynamic therapy photosensitizers are relatively small molecules and are not targeted, which causes them to accumulate in unwanted cellular tissues, such as skin, and elicit photosensitivity to sunlight following the treatment. Light sources have been guided to excite photosensitizers that are delivered to the target using nanoparticles for enhanced photodynamic effects and enhanced cell death [54]. Depending on the nanomaterial, photothermal effects can be induced upon light excitation and thereby cause local thermal coagulation and tissue necrosis without damaging the untargeted regions [55]. Light-based nanoparticle therapies are presented with two challenges: (1) the control of the temperature generated by nanoparticles upon exposure, and (2) extending light penetration due to intervening tissue absorbance, especially at wavelengths between 270–665 nm.

Among multiple types of nanoparticles, gold nanoparticles (AuNPs) have been most widely investigated in cancer research as agents to deliver photothermal effects. The shape and size of these AuNPs may be altered to shift the peak absorbance range toward the near-infrared (NIR) spectrum. Three AuNPs nanoparticle types (nanoshells, nanorods, and nanocages) display plasmonic properties and have been reported to raise temperatures by more than 20 °C when irradiated by an 805 nm laser at a power density of 0.8 W/cm^2^ for less than 10 min [56]. NIR light has distinctive potential advantages in clinical research by avoiding tissue absorbance and allowing efficient penetration through tissues [57,58].

Magnetic field. One of the advantages of using magnetic waves vs. visible and infrared radiation is that tissues rarely absorb magnetic waves, thereby allowing enhanced penetration through the body [38]. Accordingly, materials that exhibit magnetic behaviors, such as iron oxides, have been explored in nanomedicine in conjunction with external magnetic fields. These external magnetic fields can be used to direct magnetic nanoparticles to the region of interest for desired biological effects [59,60]. One of the most common uses of the superparamagnetic properties of magnetic materials and iron oxide nanoparticles (IONPs) is as contrast agents in the magnetic resonance imaging (MRI) [61,62,63]. It has been reported that approximately 35% of clinical MRI scans require contrast agents to improve the sensitivity and accuracy of the imaging [64] and as such, contrast agents based on iron oxides, such as magnetite, maghemite, or hematite, have been widely used to aid imaging and diagnosis of lesions during MRI [65,66]. The superparamagnetic properties of IONPs can also serve as carriers of pharmaceutical agents against specific diseases, thereby greatly enhancing the therapeutic index [67].

The application of hyperthermia therapy with IONPs is distinct from the photothermal therapies of AuNPs. Unlike light-based thermal therapies, alternating magnetic fields are used to stimulate IONPs, which then subsequently generate frictional heat. The heat generation of superparamagnetic IONPs depends on the strength of the magnetic field, magnetic properties, and the size of the nanoparticles [60,68]. If the diameter of IONP is smaller than 100 nm, heat is produced due to the friction of IONPs according to the magnetic field gradient. On the other hand, for larger IONPs, heat is rather generated by the rotation of the magnetic moment at each field oscillation [69,70]. As mentioned above, magnetic hyperthermia is superior to light-based approaches in that the magnetic waves may penetrate tissue better [71]. On the other hand, hyperthermia treatment, in general, requires longer and repeated exposure with a magnetic field compared to the length of laser irradiation required for photothermal ablation due to the limited power of magnetic fields.

Ultrasound. Focused ultrasound has been used for different applications, including high-intensity focused ultrasound (HIFU) techniques for focal ablation of tumors [72] and drug delivery to inaccessible regions, such as the central nervous system [73]. Focused ultrasound uses the mechanical acoustic energy generated by piezoelectric elements to cause a rise in local temperature, histotripsy with the fragmentation of tissues due to extreme high-pressure waves generated [74], and temporary disruption of cell membrane integrity [75]. One of the main advantages of focused-ultrasound mediated therapies is that their depth of ultrasound penetration is not limited as much as electromagnetic radiation, allowing access to tissues that are located in the deeper regions. Different nanoparticles have been utilized to enhance the thermal effects of focused ultrasound therapy [76] as well as to deliver drugs in nano-formulations for enhanced therapeutic index [77]. Sonosensitizers, molecules that elicit potent toxicity to cells when combined with ultrasound waves, may also be conjugated onto nanoparticles to deliver sonodynamic effects against different tumors [78].

## 3. CNPs as Theranostic Tools for Cancer

Pure carbon exists in various allotropes, with the most well-known forms of naturally occurring carbon allotropes being diamond (sp^3^ bonds) and graphite (sp^2^ bonds). The electrical, mechanical, optical, and thermal properties of carbon-based materials and their diverse allotropes have gained great interest in recent years in biomedical fields such as biosensing, drug delivery, tissue engineering, and target-specific theranostics [79,80]. Among the different CNPs, graphene, carbon nanotubes, and fullerenes (Figure 2) are three of the most commonly utilized materials in the biomedical field. Because of their high surface area consisting of sp^2^-bonded carbon atoms, carbon nanoparticles are highly hydrophobic in their pristine forms [79]. However, the surface of CNPs can be functionalized, allowing them to become soluble in aqueous solutions [81,82]. Furthermore, CNP surface functionalization allows for the application of surface chemistry options for the attachment, non-covalently or covalently, of different molecules of interest for diagnostic or therapeutic purposes.

### 3.1. Carbon Nanotubes (CNTs)

CNTs are well-ordered, hollow, carbon graphitic nanomaterials with cylindrical structures consisting of graphene sheets rolled at specific and discrete angles [86,87]. First discovered in 1991 by Sumio Iijima, CNTs have since then gained a reputation in biomedical fields based on their unique structures and properties, including strength and size (stability, high aspect ratio, and large surface area) [88,89]. Typically, CNTs are available in single-walled forms (single-walled carbon nanotube, SWCNT) or multi-walled forms (multi-walled carbon nanotubes, MWCNT, consisting of several layers of carbon sheets rolled in concentric layers) depending on the method of synthesis [90]. Both forms have been regularly used for biomedical purposes, and there are indications of which form may be more suitable for specific purposes [91]. For example, SWCNTs consisting of a single layer of rolled graphene sheet boast a large spatial surface area (up to 1300 m^2^/g in SWCNT vs. a few hundred m^2^/g in MWCNT [92]), onto which a molecule of interest may be loaded. Furthermore, due to the hollow inner structure, drugs may be incorporated into the inner cavities of CNTs allowing for the protection of unstable drugs from the biological environment and allowing for controlled drug release depending on the tube diameter [93,94]. On the other hand, MWCNTs have more defects in their structure during synthesis compared to SWCNTs, which, ironically, would make surface functionalization and modification more accessible [91].

One of the biggest challenges in the use of CNTs is their innate hydrophobicity. CNTs regularly aggregate into bundles when suspended in an aqueous environment and must be dispersed before in vivo applications. Surface modifications of CNTs by acid oxidation can introduce functional groups, such as carboxyl, hydroxyl, phenolic, and lactone groups, on their surface [95], allowing for polarization of the CNT molecule and, therefore, enhancing their solubility in aqueous solutions. If functionalized for water solubility, CNTs have been observed to clear away from circulation within a few hours with no specific organ accumulation [96]. Carboxylated SWCNTs have also been reported to degrade naturally within 90 days by phagolysosomal simulant fluid, avoiding any long-term accumulation in organs [97]. Further surface modification with PEG chains can enhance their length of staying in circulation. Longer PEG chains on the CNTs grant improved circulation times as fewer nanoparticles coated with PEG5400 are observed to be removed from circulation by the reticuloendothelial system (RES) compared to the PEG2000-coated ones [98].

The concept of CNT-mediated drug delivery to tumor targets is similar to other forms of nanoparticle drug delivery. The aromatic rings of CNTs allow for both pi stacking and hydrophobic interactions, which are two crucial non-covalent interactions essential for NP-drug binding. Similarly, CNTs can also be used as scaffolds to carry and target drugs for more enhanced therapeutic effectiveness; more often, these drugs alone have a narrow therapeutic window due to systemic toxicity [89,99]. Classes of anticancer agents that have been loaded onto CNTs for targeted delivery include topoisomerase inhibitors [100], anthracyclines, such as doxorubicin [101], platinum-based drugs (DNA chelators, such as cisplatin and carboplatin) [102], antimetabolites (disrupts metabolic pathways during the formation of nucleic acids in cancer cells, including antifolates methotrexate and purine/pyrimidine antagonists, such as 5-fluorouracil and gemcitabine) [103], and anti-microtubules [104]. Once the CNT-drug complex reaches the tumor mass via active or passive targeting, the complex is internalized, and the drug load is released by designated mechanisms [99]. Several mechanisms have also been proposed for this enhanced cellular uptake of CNTs, including energy-dependent receptor-mediated endocytosis [99,105]. On the other hand, internalization of the CNTs may also occur across the cell membrane in an energy-independent passive manner through diffusion or penetration [106,107]. Based on their small size (sub-1 μm in length) and their needle-like shape, SWCNTs may also penetrate across the membrane without energy expenditure by “piercing” the cell [108]. Accordingly, penetration into the cell membrane via such piercing processes has been suggested as a mechanism for the cytotoxicity associated with CNTs. However, further investigations are required to fully understand the effects of internalization and cytotoxicity associated with carbon-based nanomaterials.

CNTs have distinct light-absorbing properties that also make them ideal candidates as photothermal agents. First, they efficiently convert NIR radiation into a heat [107]. It has been reported that exposure of CNTs to high-energy laser leads to a dramatic increase in the temperature up to a few thousand degrees within microseconds [109]. The heat generated by CNTs is a result of electronic excitation upon laser exposure which is converted into a molecular vibrational energy [110]. Furthermore, unlike many bulk metals, CNTs also possess a very broad electromagnetic absorbance spectrum covering both NIR I and II windows ranging from 650 nm to 1350 nm, which is the ideal range for efficient tissue penetration [111,112]. During the laser exposure, the temperature profile of CNTs and its surrounding is a function of the distance relative to the radius of the CNT, with larger diameter CNTs, maximizing the laser heating process [113]. Therefore, the photophysical absorptive properties of CNTs may be engineered according to the “nano-antenna effect”, by tailoring their length and diameter (the wall number of the tube) and, by doing so, retain their unique radiation–thermal conversion efficiency required for the effective photothermal therapy [114]. The initial study on the photothermal ablation of tumors using CNTs was performed in 2005 by targeting the folate receptors in HeLa cells [115]. Confocal microscopy imaging confirmed that upon reaching the targeted HeLa cells, the functionalized CNTs were internalized, and with subsequent 5-min exposure to an 808 nm laser with a power of 3.5 W/cm^2^, irreversible damage to the tumor incurred. Since then, different strategies have been employed to further optimize CNTs’ photothermal efficiency. Cancer-specific antibodies, such as α-PSMA [116], α-TSHR [84], α-IGFR1 [117,118], or α-HER2 [118], and cell-specific α-CD22 and α-CD25 targeted antibodies [119] all have been used to achieve a more selective cell ablation effects (see Table 1).

### 3.2. Graphene

In 2004, a single layer of graphene was isolated and characterized by Andre Geim and Konstantin Novoselov at the University of Manchester using the “scotch tape” technique [138]. Graphene consists of a single layer of sp^2^-bonded carbon atoms that are packed into a unique two-dimensional honeycomb crystal lattice [86,139]. Graphene oxide (GO), a derivative of graphene, is often considered for biomedical applications due to the ease of preparation, its richness in functional groups, and extensive surface area availability [122,140]. Before the layers of graphene were discovered and isolated from graphite crystals, which were shown to have unique electrical, optical, chemical, and mechanical properties, it was merely regarded as a part of the graphite’s crystal structure [141]. The planar structure, availability of functional groups, such as epoxy bridges, hydroxyl, and carboxyl groups, and the abundance of delocalized pi electrons in GO sheets endow them with an outstanding ability to immobilize different substances, including drugs, ligands, and fluorophores for theranostics purposes [142,143]. In recent years, the properties of graphene sheets have begun to threaten the dominance of CNTs in potential biomedical applications. Graphenes, especially in the PEGylated forms, are reported to have lower in vitro cytotoxicity compared to CNTs [144].

The large surface area of graphene also allows for the surplus availability of pi electrons which can be used to form pi stacking interactions available to some aromatic anti-cancer drugs, including doxorubicin and camptothecin [145]. The loading capacity and the release of these anti-cancer drugs are found to be dependent on the hydrogen bond formed between the hydroxyl and the carboxyl group of the GO and the amine groups on the drug, as well as the pH of the environment [146,147]. Once administered and delivered to the targeted cell, there are two proposed routes of internalization: one is energy-dependent endocytosis or phagocytosis, and the other is energy-independent direct penetration into the target cell [148]. Some suggest that the GO-based nanoparticles enter cells only through energy-dependent endocytosis based on the lower cytotoxicity observed in cells by GO-based nanoparticles compared to the cytotoxicity profiles observed with CNT-based nanoparticles [149]. Lastly, targeting moieties specific to the tumors may also be engineered onto the surface of the GO-nanoparticles, promoting further specificity in drug delivery, thus minimizing potential cytotoxic side effects on non-cancerous cells.

The potential of GO and its derivatives as a photothermal agent has been evaluated in recent years. Combined with the strong NIR optical absorption capacity, efficient thermal conversion, and potential tumor-targeting specificity, GOs have become a strong candidate for such methods. The first in vivo study using PEGylated GO-nanoparticles in photothermal therapy was examined in 2010 [124]. In this study, 4T1 breast tumor xenograft mice were injected with the nanoparticles intravenously and via passive targeting, accumulated (EPR effect) at the tumor site as confirmed by fluorescent labeling. The tumor mass was irradiated with an 808 nm laser for 5 min at a power of 2 W/cm^2^. The investigators found that 100% of xenograft tumors were eliminated post-treatment, indicating the effectiveness of GO nanoparticles as in vivo photothermal agents. To complement these results, similar ultra-small reduced nano-GOs (nRGO-PEG) were prepared, in which the PEGylated GO nanoparticles were reduced again before being re-coated with PEGylated phospholipid [123,125]. The additional reduction of the GO nanoparticles “cleaned” residual functional groups from the surface, yielding a GO product with higher optical potential and enhanced NIR absorbance [140]. The nRGO-PEG was conjugated with an Arg-Gly-Asp peptide to encourage selective cellular uptake to targeted U87MG glioma cell lines. This improvement in surface modification also improved tumor ablation efficiency in vivo with the injection of nRGO-PEG into 4T1 breast cancer xenograft tumors resulting in complete ablation in 5 min with an 808 nm laser needing only 0.15 W/cm^2^ of laser power [125]. The ability to achieve a complete therapeutic effect using a very low optical power is beneficial not only in minimizing nonspecific damage of untargeted tissues but also in improving treatment efficacy when applied to larger or internal tumors.

### 3.3. Fullerene

The structure of fullerenes was theoretically predicted since the 1960s [150,151], and they were first identified (C_60_ and C_70_ buckyballs) with mass spectrometry by Harold Kroto from the University of Sussex and Richard Smalley from Rice University in 1985 [152]. Just like other carbon allotropes, such as CNTs, fullerenes are hollow carbon clusters that also consist of sp^2^ carbons that form symmetric sphere-shaped cages with various sizes [153]. One of the most common forms of fullerenes, C_60_, consists of 60 carbon atoms with 12 pentagons made of C_5_–C_5_ single bonds and 12 hexagons made of C_5_–C_6_ double bonds [153]. Unlike graphene and carbon nanotubes, the size distribution of fullerenes is relatively uniform, and the sizes of the common fullerenes in biomedical applications are often smaller than 5nm; although these nanoparticles would passively accumulate at the tumor microenvironment [154], they would also be rapidly cleared from the body by renal filtration and urinary excretion, potentially minimizing their efficacy [155].

The hollow fullerene cages can be utilized for diagnostic purposes. In certain circumstances, metal atoms can be placed within the interior of the fullerene cage, forming what is known as endohedral fullerenes or metallofullerenes [156]. Often, the minimum size for such endofullerenes is C_60_ due to the size restraint of entrapped metals, while larger fullerene molecules, such as C_80_ or even larger, are preferred to enclose metals, such as lanthanum, yttrium, scandium, and gadolinium [157]. Accordingly, gadolinium-enclosed [158,159] or conjugated [160] fullerenes have been evaluated as MRI contrast agents for cancer diagnostics. In addition, radioisotopes, such as ^177^Lu radionuclide-encapsulated fullerenes, have been examined for both radiodiagnostic and therapeutic purposes [161].

Just as fullerenes were used for diagnostic purposes, different variations have also been designed for drug-delivery-based cancer therapies. Because of their small sizes, fullerenes can readily cross the cellular membranes without damaging cells if they are properly functionalized to be soluble in aqueous environments [162]. This internalization property makes fullerenes an optimal vehicle for carrying chemotherapeutic agents into the tumorigenic cells allowing a great increase in therapeutic efficacy. Various chemotherapeutic payloads loaded onto fullerenes have been evaluated for in vitro and in vivo efficacy, including doxorubicin [163], paclitaxel [164], cisplatin [165], and even DNA-based gene therapies [166].

While photoacoustic images could be obtained using high-contrast polyhydroxy fullerenes, photothermal effects are also observed when injected directly into the tumor with 30 μL (0.45 mg/mL) of the fullerene nanoparticles and irradiated immediately with a 785 nm laser at 0.5 W for 10 min [121]. Different variations of fullerenes have also been evaluated for their effectiveness in photodynamic therapy as well. Known for their photostability, C_60_ and C_70_ fullerene variations have been reported for potential applications in photodynamic therapies [167,168]. In one study, 0~20 μM of C_60_ fullerenes were incubated with CCRF-CEM acute lymphocytic leukemia (ALL) cells for 24 h in vitro. Light wavelengths ranging from 365 nm to 650 nm were applied, and it was seen that at 365 nm, 4 J/cm^2^ of light was sufficient to induce over 90% of cell death, while for light at higher wavelengths (515 nm and 650 nm) no significant cell death was observed even at 20 J/cm^2^ [167]. In another study, photodynamic treatment using a C_70_ fullerene formulation was carried out on HeLa-Luc xenografted tumors treated with a 980nm laser at 0.7 W/cm^2^ for 3 min where the authors did not observe any tumor recurrence 14 days following phototherapy [169]. However, the main absorption of light photons by fullerenes occurs in the ultraviolet range, where the high energy levels could induce irreversible non-specific tissue damage as well as be limited by the penetration of the tissue depth [167,170]. Such limitations can be overcome by using additional photosensitive molecules, as shown by Guan et al., where a C_70_ fullerene-based platform was conjugated with chlorin e6 photosensitizers [168]; another method is to apply the principles of two-photon excitation to match the required energy level using lower energy intensity [170].

## 4. Potential Translation of Nanoparticle Theranostics into the Clinic

We have discussed the theranostic applications and research investigations of different nanomaterial platforms in in vitro and in pre-clinical in vivo settings. Nevertheless, the ultimate objective of these studies would be to translate their innovativeness and assess their utility in a clinical setting, thereby providing physicians with alternative therapeutic options and, possibly, a new standard of care. Over the last few years, the unique photoablative properties of the abovementioned nanoparticles have been applied for possible therapeutic purposes. Their theranostic potential is expanded by labeling the nanoparticles with either contrast or radioactive agents, thereby allowing physicians to identify and obtain a better understanding of the tumor and the treatment region.

However, cancer is a heterogeneous disease with extensive tumor heterogeneity encompassing variabilities in genomic sequence, gene expression, metabolism, motility, proliferation, and metastatic potential [6,171]. These variations may occur between different tumor masses (inter-tumoral) [7,8] and/or within a single tumor mass (intra-tumoral) [7,8,9]. With the expansive use of next-generation sequencing technologies, the extent of somatic DNA alterations in tumors is now substantiated, with the mutational landscape of cancers being greater than what was originally anticipated [172]. Another revelation of tumor sequencing in cancers is the presence of excessive mutations between tumor foci vs. adjacent tissue samples that are histologically “normal” [173]. Multifocal tumors, such as prostate cancer, are often identified to be genetically distinct from each other. Nevertheless, intra-tumor heterogeneity of non-driving somatic mutations common in both malignant and normal tissue samples can complicate the use of biologically targeted treatments, as the characterization of the dominant biology of patients’ cancer becomes difficult. With individual tumor cells developing functionally distinctive and critical pathophysiological features that would promote cancer progression, overcoming the almost innate genetic heterogeneity of tumors becomes the principal challenge in the cancer treatment [9].

The phenomena of cancer recurrence to therapeutic interventions are often observed with prolonged treatment with biological agents [174]. Biological agents place selective pressure on tissues, resulting in cells with initial resistance to proliferate and eventually leading to a new malignant mass. Examples of selectively targeted mutations to therapeutic agents are anti-androgens in the treatment of prostate cancer, where the androgen receptor incurs a mutation to the targeting ligand binding domain, where now the antagonizing drug functions act as agonists [175,176]. Mutations in the androgen receptor have been found to circumvent antiandrogens, such as bicalutamide, enzalutamide, and apalutamide [177,178,179]. In advanced prostate cancer and castrate-resistant disease state, a class of mutations in the androgen receptor has been identified with a complete loss of the ligand binding domain and functionally constitutively active without any need for androgen ligands [180]. Hormone ligand-directed therapies, such as enzalutamide and abiraterone, in this scenario, would be completely infective. These scenarios present a “tug of war” between the development of next-generation therapeutic modalities and the expansion of new malignant tumors.

A critical discussion point for the use of nanoparticles and their application in focal therapy is their ability to deliver tumor-destructive effects to the targeted area with minimizing the damage to nearby tissue or incurring non-specific/off-target effects. The selection of CNTs, functionalized with tumor-targeting moieties, offers significant advantages and benefits over other inorganic nanoparticles, and delivers the medical needs unmet by the current cancer therapeutics:(1)The combination of intravenous administration of a targeted CNT formulation with non-invasive focal photothermal therapy will result in tumor ablation with minimal tissue damage. This is a result of the accumulation of a critical mass of nanomaterial, and the amount of heat generated upon light stimulation will confer total tumor ablation even if not every cell has been targeted. This is a positive consequence of treatment, as the ablation has a “near-field” effect on tumor cells and the adjacent surrounding cells. As these adjacent cells are influenced by the tumorigenic properties of the nearby neoplasia, ensuring expansion of the ablation field is important as they have the potential to undergo neoplastic transformation.(2)The ability to target CNTs to all tumor cells independent of their genetic profiles, the CNT/photothermal effects will be limited to cells expressing the tumor-specific antigen. However, should tumors be proven to be resistant to the targeting moiety, there is the possibility of personalizing and adjusting/adding the targeting moiety against different surface antigens. The adaptability of tumor target agent selection will come from new emerging information that is garnered from biomarker evaluations and unique mutation profiles defined from patient tumor sequencing.(3)Complete and rapid tumor ablation should not afford the selection of resistant tumors, as is commonly observed with prolonged biological therapeutics. The photoablation of the tumor would be total after the first and single treatment, with the likelihood of tumor recurrence significantly reduced, unlike recurrent chemotherapy or radiation therapy. The procedural safety inferred from using targeted CNT treatment is substantive, considering the consequential side effects resulting from existing therapies [181].(4)The ability to accurately diagnose and monitor disease onset, recurrence, and progression is critical. Targeted CNTs can provide tumor diagnostic and surveillance tools, as it comes from the ability of dual labeling with an agent that is compatible with existing imaging platforms, i.e., MRI, PET, or even ultrasound.(5)The overall costs of cancer therapeutic drugs and other treatment options are typically expensive. Although CNTs are much cheaper to manufacture than most nanomaterials, the process of functionalizing CNTs with both the targeting and imaging moieties brings added costs to the final formulation. However, therapeutic CNT platforms should not contribute to additional hospital costs and stay times in the same capacity as surgery and radiation interventions. Altogether, functionalized CNTs as a photothermal treatment protocol can become a standard of care for patients with early-stage and localized disease, with the new treatment being safer and demonstrating a clear improvement in overall survival benefit.

Clinical applications of CNPs have been limited, predominantly as contrast tracers during surgical applications, and only a handful of clinical trials have been conducted to evaluate their clinical utility with two trials evaluating the imaging potential of CNTs (NCT01773850 and NCT04495634). However, the current clinical landscape of inorganic-based nanoparticles is expanding. IONPs as contrast agents are being used to improve the quality of MRI (ClinicalTrials.gov Identifier: NCT01895829, NCT02744248, NCT02511028, NCT01815333). Nevertheless, it remains to be seen whether these nanoparticles are approved as previously available iron oxide contrast agents, such as ferumoxides (Feridex) and ferumoxsil (Gastromark), have been discontinued from clinical use in the U.S. due to regulatory and marketing concerns [182]. On the other hand, several ongoing clinical trials are using a variety of nanomaterials as either imaging or therapeutic agents. Table 2 outlines promising companies using inorganic or synthetic polymer-based nanoparticles that are in or entering clinical trials. These nanotherapeutic platforms are serving as either drug/genetic payload carriers (Bind Therapeutics and New Link Genetics) or physical enhancing agents (Nanobiotix and Nanospectra Biosciences).

The concept of improving the therapeutic index by delivering higher concentrations to or near the tumor mass has been in practice for a while in the clinical landscape. One example of nanomaterial-mediated chemotherapy is ABI-007 or Abraxane^®^, where the protein albumin is a delivery vehicle for paclitaxel [183]. While paclitaxel itself is a highly proficient anti-cancer agent, it is poorly soluble in water due to its structural hydrophobicity and initially required the use of solvents, such as Cremophor EL and ethanol, for clinical uses despite the severe allergic reactions associated with them [183,184]. Therefore, albumin is used to carry the water-insoluble paclitaxel in circulation. Numerous clinical trials using these albumin-bound paclitaxel formulations were performed to confirm the superior therapeutic efficacy [185,186]. In response to the success of ABI-007, nanoparticle formulations were developed consisting of synthetic polymers, with New Link Genetics developing CRLX101 (cyclodextrin-based polymer carrying camptothecin) [187] and Bind Therapeutics introducing BIND-014 (PSMA targeting Accurin polymeric nanoparticle containing doxorubicin) [188]. CRLX101 and BIND-014 have been assessed in a number of clinical trials for different malignancies. Similar to traditional drugs, both platforms underwent multiple dosing. BIND-014 was well-tolerated in safety studies, with 30% of the 42 prostate cancer patient in the trial showing a ≥50% decline in PSA levels [189]. However, it has not progressed beyond Phase 2 trials. In 2016 BIND Therapeutics was purchased by Pfizer, and in 2019, New Link Genetics merged with Lumos Pharma. Nonetheless, there are no indications from existing company profiles of continuing clinical trials with either nanoparticle-drug formulations at the moment.

On the other hand, the progression of clinical trials with nanoparticles as physical-directed or enhancing therapeutic platforms has been promising. Nanospectra Bioscience is leading clinical trials to assess nano-directed photothermal therapy. Using gold nanoshells (Aurolase™), an initial non-randomized pilot study regarding safety and efficacy was performed on 16 patients with early-stage prostate cancer [190]. All patients with early prostate cancer were recruited and treated with continuous near-infrared wavelength laser light. Post-treatment biopsies showed residual tumors in four patients at a 3-month follow-up, and two men had biopsy-proven cancer at a 12-month follow-up. Still, the procedure itself demonstrated that there was no evidence of any side effects experienced (even after one year) by patients in urological, rectal, and other general indices. The trial also provided information on the efficacy of nanoparticles in administering focal ablation with the sparing of damage to normal nearby tissue. These results are very encouraging in addressing potential safety issues of inorganic nanoparticles for use in biomedical therapeutics. A recent clinical trial of 30 men (NCT04240639) would provide enough positive data for Nanospectra to consider FDA filing for their device [191]. However, Nanospectra’s platform is not without its limitations. An outstanding concern is that the company employs an untargeted nanoparticle that depends upon passive accumulation at the tumor site. Therefore, successful tumor ablation will depend upon the sufficient concentration of the nanoshells in the tumor space. A second limiting caveat is that the presence of the nanoshells at the tumor site is transient and not adhered to in place, which could limit the treatment window from achieving maximal tumor ablation. Nanobiotix has developed NBTXR3, hafnium oxide-based nanoparticles, used to enhance and amplify the energy deposit within tumors in the radiotherapy [192]. It has been studied in the soft tissue sarcoma [193], head and neck squamous cell carcinoma [194], and hepatocellular carcinoma [195]. For soft tissue sarcoma, NBTXR3 had a 16.1% vs. 7.9% (radiotherapy alone) pathological complete response rate (*p* = 0.04) with an overall safety profile of NBTXR3 similar to those receiving radiation only. Recent evidence suggests that NBTXR3 has been shown to stimulate an immune response that is not observed with radiotherapy alone, with an increase of CD8+ T cell infiltration [196]. Similarly, NBTXR3 is also non-targeted and is injected intratumorally to maximize tumoral accumulation.

## 5. Conclusion Remarks

Modern methods in medicine have revolutionized our management of neoplastic disorders. Our increased understanding of the underlying mechanisms of cancer will provide us with continuing potential therapeutic avenues. Perhaps the single biggest challenge in the management of cancer today is the inevitable development of resistance phenotypes exerted by current therapeutic regimes creating selective pressures [197]. The application of nanomaterials has been added to the cancer research therapeutic armamentarium. Inorganic and carbon-based nanoparticles, as immediate agents, can deliver or enhance physical energy at the site of the tumors. In doing so, the cells are not susceptible to selective genetic pressures and leave irreversible damage. Photothermal-based pre-clinical animal models and clinical trials suggest these modalities confer positive outcomes and will be a viable option in cancer therapy.

As nanoparticles are currently proposed as therapeutic agents, their toxicity has been investigated with numerous clinical studies assessing their safety profile, and more studies will be needed. The FDA has officially declared nanoparticles to be neither harmful nor safe and will assess the safety profile on a per-nanoparticle basis. They have provided a guidance report for the industry and for the application of nanoparticles [FDA-2010-D-0530 (accessed on 8 June 2018)] The FDA is willing to support and define the regulatory process related to therapeutic nanoparticles. The National Cancer Institute has also established an NCI Alliance for Nanotechnology for Cancer. Furthermore, their discussions on nanoparticle theranostics have also been updated, with all NCI alliance sites working hand-in-hand with the FDA regarding the safety profiles of nanoparticles.

The FDA will consider the current framework for safety assessment sufficiently robust and flexible to be appropriate for a variety of materials, including nanomaterials, and maintain a product-focused, science-based regulatory policy.Technical assessments will be product-specific, and this will consider the effects of nanomaterials in the biological and mode of action (e.g., photothermal) context of each product and its intended use.As such, the policies for each product area, both substantive and procedural, will vary according to the statutory authorities and relevant regulatory frameworks. This regulatory policy allows for tailored approaches that adhere to applicable legal frameworks and reflect the characteristics of specific products or product classes and evolving technology and scientific understanding.Moreover, the industry remains responsible for ensuring that its products meet all applicable legal requirements, including standards for safety, regardless of the emerging nature of the technology involved in the manufacturing of a product. The FDA also encourages the industry to consult with the Agency early in the product development process to address any questions related to the safety, effectiveness, or other attributes of products that contain nanomaterials or about the regulatory status of such products. Early consultations with the FDA will facilitate a mutual understanding of the specific scientific and regulatory issues for nanotechnology products.

A major distinction needs to be made for the classification of nanoparticles used in conjunction with external stimuli, such as photo-ablative therapy or radiation, and their classification as medical devices. This is because the external stimuli will activate the intrinsic physical properties of the particles rather than being dependent on modulating a cellular pathway, thereby eliciting a process of programmed cell death. Therefore, the application of nanotherapeutic platforms will result in product attributes that differ from those of conventionally-manufactured products (i.e., biologicals) and, thus, may merit a different examination and classifications for clinical trials process vs. investigational new drugs.

## Figures and Tables

**Figure 1 bioengineering-10-00108-f001:**
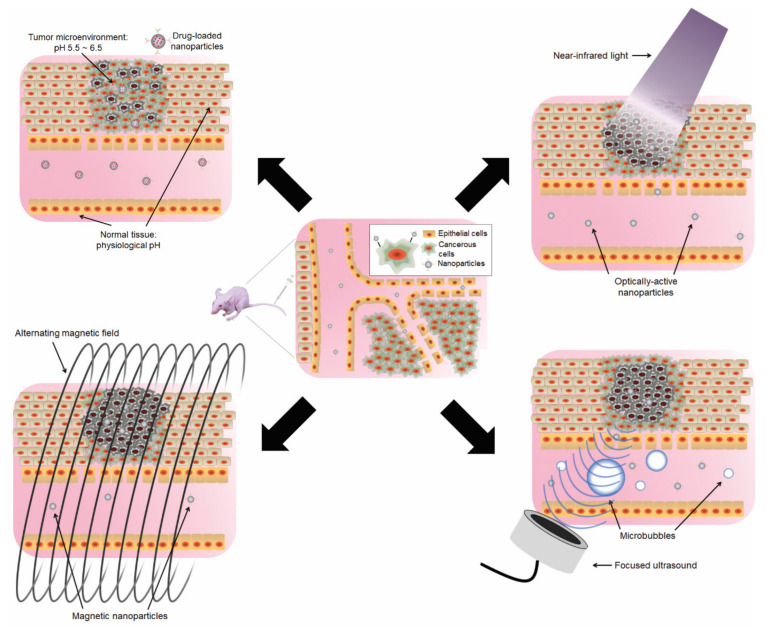
The use of nanoparticles and energy to maximize the therapeutic index. In the tumor microenvironment, the integrity of the blood vessel wall lining is undermined due to uncontrolled vasculature growth, allowing increased accumulation of molecules. The use of nanoformulations to prolong the half-life and to exploit such enhanced permeability and retention (EPR) effect is central in the field of nanomedicine. pH: The tumor microenvironment is slightly more acidic (pH 5.5~6.5) than normal tissues, allowing the selective release of therapeutic cargos from the nanoparticle carriers. NIR Light: The use of near-infrared light (>800 nm) to excite nanoparticles for imaging and photodynamic/photothermal therapies. Magnetic Field: Magnetic fields are applied to the tumor to guide superparamagnetic nanoparticles toward the region of interest, as well as for magnetic resonance imaging. Ultrasound: The combination of focused ultrasound and microbubbles can assist in nanoparticle delivery and tumor ablation (HIFU thermal ablation and sonosensitization).

**Figure 2 bioengineering-10-00108-f002:**
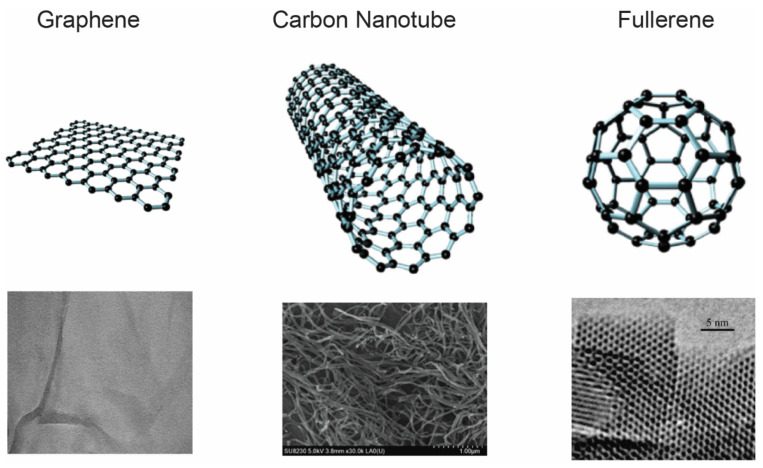
Carbon nanoparticles. 2D diagrams and electron microscope pictures of graphene, carbon nanotubes, and fullerene C_60_. Parts of the electron microscope picture of graphene are reproduced from [83], the electron microscope picture of carbon nanotubes is reproduced from [84], and the electron microscope picture of fullerene is reproduced from [85]. Copyright permissions were obtained from the authors for the republishing of the figures.

**Table 1 bioengineering-10-00108-t001:** Comparative table of nanoparticles for photo-ablation listed in this review ^1^.

Study	Type of Nanoparticles	Indication	Target	Amount of Nanoparticles Used	Laser Wavelength (nm)	Laser Power (W/cm^2^)	Time	In Vitro/Vivo (Route) ^2^	Results and Comments
**Carbon-based**									
Chakravarty et al. (2008) [119]	SWCNT	Daudi (Burkitt’s lymphoma)	CD22, CD25	40 μL, 0.09 mg/mL	808	5	7 min	in vitro	Specific in vitro photothermal cell ablation by antibody-directed methods. Antibody-bound CNTs remained with cells in serum solution despite washing with PBS.
Dotan et al. (2015) [84]	MWCNT	BCPAP (thyroid)	TSHR	n/a	532	2.7	30 s	in vitro	In vitro evaluation of MWCNT-mediated cell ablation. ~73% cell ablation achieved.
Kam et al. (2005) [115]	SWCNT	HeLa (cervical)	Folic acid	Up to 5 mg/L	808	1.4	2 min	in vitro	In vitro evaluation of SWCNT endocytosis and CNT-induced photothermal ablation.
Khan et al. (2012) [120]	Gold nanocage-SWCNT	LNCaP (prostate)	A9 RNA aptamer targeting PSMA	3 nM	1064	2	10 min	in vitro	In vitro photothermal ablation of LNCaP cells using GNC-CNT hybrid resulted in over 95% of cell death.
Krishna et al. (2010) [121]	Fullerenes	BT474 (breast)	No	30 μL of 0.45 mg/mL	785	0.5	10 min	in vivo (i.t.)	Histological sections of tumors obtained 20 h post-treatment showed up to 40% necrosis in the nanoparticle-treated mice, compared to 20~50% necrosis in the control.
Lee et al. (2017) [116]	MWCNT	LNCaP (prostate)	PSMA	75 μL of 60 μg/mL	532	2.7	30 s	in vitro	>65% in vitro cell ablation. The temperature of the bulk cell-CNT solution remains relatively consistent after laser exposure.
Li et al. (2014) [122]	Reduced graphene oxide	AGS (gastric cancer)	Transferrin	50 μg/mL	800	Up to 0.03	Up to 45 raster scans	in vitro	Femtosecond lasers are used to evaluate in vitro photothermal therapy of graphene oxides.
Lu et al. (2019) [117]	SWCNT	BXPC-3 (pancreas)	IGF1-R	300 μg/mL, 200 μL	785	1	5 min	in vitro/in vivo (i.v.)	SWCNT-based, image-guided photothermal therapy on an orthotopic murine model of pancreatic cancer-bearing BXPC-3.
Markovic et al. (2011) [86]	Graphene and SWCNT	U251 (glioma)	No	Up to 10 μg/mL	808	2, spot size 6 × 8 mm	Up to 5 min	in vitro	Graphene functionalized by PVP, CNT by DNA. In vitro IC_50_ of 0.3 ± 0.0 μM and 4.2 ± 0.3 μM for graphene and CNT respectively when irradiated for 5 min.
Robinson et al. (2010) [111]	SWCNT	4T1 (murine breast)	No	200 μL, 0.35 mg/mL	808	0.6	5 min	in vivo (i.v.)	Compared the photothermal efficiency of SWCNT vs gold nanorods in vivo. GNR required higher power (2 W/cm^2^) to achieve complete tumor remission than CNT (0.6 W/cm^2^)
Robinson et al. (2011) [123]	Reduced graphene oxide	U87MG (glioma)	RGD peptide	6.6 mg/L	808	15.3	8 min	in vitro	Destruction of U87MG cells when incubated with RGD peptide-functionalized reduce graphene oxide in vitro.
Shao et al. (2007) [118]	SWCNT	BT474, MCF7 (breast)	IGF1R, HER2	4 μL, 0.1 mg/mL	808	0.8	3 min	in vitro	Two different functionalization—one with α-IGF1R antibodies, and another with α-HER2 antibodies on different sets of CNTs were prepared, incubated with the cells, and irradiated with laser in vitro.
Torti et al. (2007) [114]	Nitrogen-doped MWCNT	CRL1932 (kidney)	No	Up to 0.083 mg/mL	1064	3	4 min	in vitro	Evaluated the correlation of the length of MWCNT and their photothermal effects. Longer (1100 nm) CNTs were better in cell ablation (up to 95%) than shorter (300 nm) CNT in vitro.
Yang et al. (2010) [124]	Nanographene sheets	4T1 (murine breast)	No	200 μL, 20 mg/kg	808	2	5 min	in vivo (i.v.)	PEG-functionalized graphene sheets, passive targeting. Complete remission of tumor after laser irradiation in vivo.
Yang et al. (2012) [125]	Reduced graphene oxide	4T1 (murine breast)	No	200 μL, 20 mg/kg	808	0.15	5 min	in vivo (i.v.)	PEG-functionalized graphene sheets, passive targeting. Complete remission of the tumor using reduced graphene oxide and laser irradiation in vivo. Non-reduced graphene oxides did not induce photothermal effects in mouse models.
**Gold-based**									
Chen et al. (2007) [126]	Gold nanocage	SK-BR-3 (breast)	HER2	Not specified	810	1.8	5 min	in vitro	In vitro cytotoxicity was evaluated with calcein AM and EthD-1 assay; a laser power density of 1.5 W/cm^2^ was sufficient to induce a significant decrease in cell viability.
Gormley et al. (2011) [127]	Gold nanorod	S-180 (murine sarcoma)	No	200 μL, 9.6 mg/kg, OD = 120	808	1.6	10 min	in vivo (i.v.)	In vivo measurement of the internal temperature of the tumor, ∆T of 13.7 ± 2.9 °C after 10 min of exposure.
Hao et al. (2015) [128]	Gold nanoshell-deposited PLGA nanoparticle	U87MG (glioma)	Angiopep-2 peptide	Up to 1.2 mg/mL	808	Not specified	5 min	in vitro/in vivo (i.v.)	Loaded with docetaxel; the combination of photothermal effects + DTX inhibited up to 70% of cell growth in vitro, and up to 70% inhibition of tumor growth in vivo.
				Not specified		1.5	1.5 min		
Hirsch et al. (2003) [129]	Gold-silica nanoshell	SK-BR-3 (breast)	No	Not specified	820	35	7 min	in vitro/ in vivo (i.s.)	Temperature profiles were monitored with MRI. In vivo temperature increase of 37.4 ± 6.6 °C on 4–6 min NIR exposure.
		TVT (canine venereal)				4, 5mm spot diameter	<6 min		
Huang et al. (2006) [130]	Gold nanorod	HOC313 clone 8, HSC3 (oral SCC)	EGFR	OD_800nm_ = 0.5	800	10, 1mm spot diameter	4 min	in vitro	In vitro cell viability was evaluated by trypan blue staining. At 20 W/cm^2^ exposure for 4 min, normal untargeted HaCat cells were also injured.
Liao et al. (2015) [55]	Gold nanorod-loaded PCL polymersome	C26 (murine colon)	No	200 μL, 50 mg/kg	808	2.5	5 min	in vitro/ in vivo (i.v.)	Loaded with doxorubicin; hyperthermia-triggered release of DOX. Up to 73% in vitro cell ablation with a combination of GNR-DOX and laser, and the same protocol evaluated in vivo caused tumor cell necrosis and the complete removal of the tumor.
Loo et al. (2005) [20]	Gold nanoshell	SK-BR-3 (breast)	HER2	100 μL of 3 × 10^9^/mL	820	8 × 10^−7^	7 min	in vitro	In vitro cytotoxicity was evaluated with calcein fluorescence.
Melancon et al. (2008) [18]	Hollow gold nanoshell	A431 (skin)	EGFR	100 μL of 7.3 × 10^10^/mL	808	40	5 min	in vitro/in vivo (i.v.)	In vitro cell death was confirmed with EthD-1 staining. Enhanced tumor accumulation with antibody targeting compared to non-targeting GNS (not significant), confirmed using ^111^InCl_3_ radiolabeling.
Piao et al. (2016) [131]	Gold nanocage	HeLa (cervical)	No	10 μg/mL	850	0.4	5 min	in vitro/ in vivo (i.v.)	Loaded with cTL peptide for additional cell membrane disruption. Applying laser to bare GNCs did not cause cell damage in vitro, but the addition of cTL peptide did (cTL mediated cell death,~60%). Compared 0.4 W/cm^2^ (no skin damage) vs 1.0 W/cm^2^ (skin damage) in vivo with cTL-GNC.
		4T1 (murine breast)		100 μL, 4 mg/mL			10 min		
Sun et al. (2014) [132]	Gold nanocage	MDA-MB-435 (breast)	SV119 targeting sigma-2 receptor	0.2 nM	808	0.8	20 min	in vitro	Loaded with doxorubicin; 80% reduction of cell growth in vitro with laser-induced photothermal + DOX treatment, while ~25% of cell growth without laser.
Topete et al. (2014) [133]	Gold nanoshell	HeLa (cervical)	Folic acid	Up to 13 μM	808	2.5	5 min	in vitro	In vitro data; loaded with doxorubicin and SPION; hyperthermia-triggered release of doxorubicin, magnetic guidance to the target using SPION.
Topete et al. (2014) [134]	Branched gold nanoshell	MDA-MB-231 (breast)	Folic acid	Not specified	808	2	10 min	in vitro	Functionalized with human serum albumin and indocyanine green, loaded with doxorubicin for light-triggered release. In vitro: >90% cell death with DOX loaded, albumin/ICG/folic acid-functionalized gold nanoshell.
**Magnetic**									
Chu et al. (2013) [135]	Iron oxide nanoparticles	Eca-109 (esophagus)	No	100 μL, Up to 0.5 mg/mL	808	0.25	20 min	in vitro/ in vivo (i.t.)	Iron oxide nanoparticles of different shapes (spherical, hexagonal, and wire-like shapes) were evaluated; 50% of cell death post laser irradiation In vitro, and significant tumor growth inhibition with continuous laser treatment after nanoparticle administration in vivo.
				70 μL, 8 mg/mL			20 min/day, 24 days		
Espinosa et al. (2016) [136]	Iron oxide nanocubes	SKOV3 (ovarian), PC3 (prostate), A431 (skin)	No	[Fe] of 0.7 mg/mL, of the nanocage itself not specified	808	0.3, 0.8	5 min	in vitro/ in vivo (i.t.)	In vitro results show that for SKOV3 cells, laser irradiation resulted in 64% cell ablation while for dual laser and magnetic hyperthermia combination over 85% of cell death was observed. Similar patterns were observed for in vivo experiments—dual protocols for tumor ablation were most efficient than either laser or magnetic by themselves.
		A431		50 μL, 14 mg/mL		0.3	10 min		
Fu et al. (2020) [67]	TiS_2_ nanosheet anchored iron oxide nanoparticles	4T1 (murine breast)	No	Up to 100 μg/mL	808/1064	0.3/1.0	5 min	in vitro/ in vivo (i.v.)	Used magnets to pull the nanoparticles toward the target. A significant difference between targeting and non-targeting groups in vitro was observed when 50 μg/mL concentration was used (<5% cell viability). Tumor weight was 30% of the control with photothermal ablation in vivo and further enhanced when combined with immune checkpoint inhibitors.
					1064	1			
Shen et al. (2013) [137]	Chitosan-modified iron oxide	KB (oral SCC), MCF-7 (breast)	No	500 μL, Up to 300 μg/mL	808	2	Up to 3 min	in vitro/ in vivo (i.v.)	Cell viability dropped below 10% in vitro when the nanoparticle concentration of 75 μg/mL was incubated and irradiated with cells for 3 min. Low toxicity associated with the chitosan-modified iron oxide in vivo, complete tumor remission after the combination of nanoparticle administration and laser treatment.
		S180 (murine ascites)		200 μL, 10 mg/mL		1.5	5 min		

^1^ If a study is divided, the top row represents in vitro data, and the bottom represents in vivo data. ^2^ Route. i.s. = interstitial injection; i.t. = intratumoral injection; i.v. = intravenous injection (tail-vein).

**Table 2 bioengineering-10-00108-t002:** Therapeutic Nanoparticles Formulations in Clinical Trials.

Company	Drug	Nanostructure	Mode of Action	Clinical Trial ID/Study Indication ^1^/Cancer Type
Nanobiotix	NBTXR3	hafnium oxide (HfO2) nanoparticles	Radioenhancer	NCT04484909/&/Pancreatic cancer
				NCT04505267/&/Non-small cell lung cancer
				NCT02721056/**/Liver cancer
				NCT04862455/&/HNSCC
				NCT04615013/&/Esophageal cancer
				NCT03589339/*/Multiple advanced cancer
				NCT02805894/**/Prostate cancer
				NCT01946867/&/Locally advanced SSC
				NCT01433068/*/Soft Tissue Sarcoma
				NCT02379845/*/Soft Tissue Sarcoma
				NCT04892173/&*/HNSCC
				NCT02901483/%/HNSCC
				NCT02465593/%/Rectal Cancer
				NCT04834349/&*/HNSCC
				NCT05039632/&*/Lung and Liver metastasis
NewLink Genetics (Lumos Pharma)	CRLX101	cyclodextrin-based polymer	Drug payload delivery (camptothecin)	NCT02389985/%/Ovarian cancer
				NCT02648711/%/Solid tumors
				NCT01612546/**/Esophageal cancer
				NCT01380769/**/Non-small cell lung cancer
				NCT02187302/*/Metastatic renal cell carcinoma
				NCT03531827/%*/CRPC
				NCT01803269/%*/Solid tumor
				NCT00333502/*/Solid tumors
				NCT02010567/%*/Rectal cancer
				NCT01625936/*/Renal cell carcinoma
				NCT00753740/W/Ovarian Cancer
				NCT01652079/*/Ovarian cancer, fallopian tube cancer
**Ensysce Biosciences**		SWCNTs	siRNA and Drug payload delivery (taxol/doxorubicin)	Unknown pre-clinical stage
Bind Therapeutics (Pfizer)	BIND-014	Accurin polymer	Drug payload delivery (Docetaxel)	NCT01812746/*/CRPC
				NCT01792479/*/Non-small cell lung cancer
				NCT01300533/*/Metastatic cancers, solid tumors
				NCT02283320/*/Squamous cell non-small lung cancer
				NCT02479178/%/multiple cancers
Nanospectra Biosciences Inc.	AuroLase	Gold nanoshells	Laser assisted ablation	NCT01679470/%/Primary or metastatic lung tumors
				NCT00848042/**/Head and neck cancer
				NCT04240639/&*/Prostate cancer
				NCT02680535/*/Prostate cancer

^1^ Study Indication. *—Study Completed; **—Study Completed—has results; %—Study Terminated; %*—Study Terminated—has results; &—Recruiting; &*—Not Yet Recruiting; W—Withdrawn.

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
