# Peer review of "Functionalized Carbon Nanoparticles as Theranostic Agents and Their Future Clinical Utility in Oncology"

_bioengineering, 2023, doi:10.3390/bioengineering10010108_

Round 1
Reviewer 1 Report
The manuscript summarizes an overview of therapeutic (an to some extent diagnostic) applications of carbon nanoparticles, with a focus on tumor. The review is well written and I would recommend publication provided that some minor issues are solved.
1) the first part of the review is excessively detailed in providing general and well known information on nanoparticles as therapeutics. Since this is a well known topic, I would recommend shortening this introductory part
2) I would expect a larger space devoted to functionalization strategies for carbon nanoparticles which, as the authors discuss, are important in providing targeted systems for more tailored treatment.
Author Response
The manuscript summarizes an overview of therapeutic (and to some extent diagnostic) applications of carbon nanoparticles with a focus on tumor. The review is well written and I would recommend publication provided that some minor issues are solved.
1) the first part of the review is excessively detailed in providing general and well known information on nanoparticles as therapeutics. Since this is a well known topic, I would recommend shortening this introductory part
We have evaluated the recommendation from the reviewer to shorten the introduction. However, upon further consideration and additional revisions suggested by another reviewer, we have opted to maintain the structure (and subsequent length) of the introduction. The initial details presented in the introduction provide the needed context for the chapters that follow.
2) I would expect a larger space devoted to functionalization strategies for carbon nanoparticles which, as the authors discuss, are important in providing targeted systems for more tailored treatment.
Table 1 provides an extensive listing of different functionalization strategies undertaken on carbon nanoparticles for tailored targeting and treatment. It would have been difficult to provide individual write-ups for each. However, from what is clear, the majority of targeting strategies have used antibody targeting. Moreover, the Table does provide some commentary of the work, that could not be fitted into the body of the manuscript.
Reviewer 2 Report
The authors attempt to review the current state of cancer therapy using carbon nanoparticles (CNPs), from basic research to clinical trials. While the information on CNPs is sufficient and reflects the title, this review also includes considerable information on nanomaterials other than CNPs.
The reviewer would like to suggest the following alternatives:
1. Change the title to include information on nanomaterials other than CNPs. In this case, Table 1 should contain information on CNPs only, and other nanoparticles should be presented as a separate table in Chapter 4.
2. Reduce the contents of Chapter 4 to those related to CNPs alone. In that case, Table 1 should only contain information on CNPs, and all other information should be deleted.
As a reviewer, I strongly recommend alternative 1 given the abundance of useful information on this topic, but will defer to the authors for the final decision.
Author Response
The authors attempt to review the current state of cancer therapy using carbon nanoparticles (CNPs), from basic research to clinical trials. While the information on CNPs is sufficient and reflects the title, this review also includes considerable information on nanomaterials other than CNPs.
The reviewer would like to suggest the following alternatives:
- Change the title to include information on nanomaterials other than CNPs. In this case, Table 1 should contain information on CNPs only, and other nanoparticles should be presented as a separate table in Chapter 4.
We have placed the focus of our review on the application of different CNP formulations as therapeutic and diagnostic platforms for neoplastic disease, with an extensive discussion provided in Chapter 3. The chemistry and biology utilized by CNPs to deliver their theranostic action is not exclusive from that employed by other nanoparticles (i.e., gold or iron). Although, the reviewer provides an good argument for splitting Table 1, into two tables; one exclusive to CNP and the other to other inorganic-based nanoparticles. But along that rationale, it would require extensive restructuring of the manuscript, therefore, we referenced Table 1 sooner in the manuscript.
- Reduce the contents of Chapter 4 to those related to CNPs alone. In that case, Table 1 should only contain information on CNPs, and all other information should be deleted.
At the moment, there are few examples of CNPs that have entered clinical trials, as such, we have provided some details on those clinical trials. However, Chapter 4, presents a review of possible advantages CNPs may have over other examples of synthetic-based nanoparticles, when they are ready to enter clinical trials. Examples of physical-directed or -enhancing nanoparticles, such as those employed by Nanospectra and Nanobiotix providing positive results are included. How would CNPs be able to compete in the same sphere? Thus, we structured Chapter 4 and the clinical discussion as such to inclusively present the clinical prospect of CNPs.
As a reviewer, I strongly recommend alternative 1 given the abundance of useful information on this topic, but will defer to the authors for the final decision
Reviewer 3 Report
The paragraph "However, in recent years, the utilization of nanoparticles has extended beyond the 40classical drug delivery methods to exploit their innate properties. Such unique properties 41of nanomaterials include but are not limited to, utilizing their high surface area to volume 42ratio for enhanced cargo loading, enhanced magnetic properties, and unique optical char- 43acteristics, which may be further augmented by external stimuli such as pH, temperature, 44light, and magnetic fields." needs references. The reviewer recommends: DOI:10.1021/acsami.1c02338 and DOI: 10.3390/pharmaceutics13030416
In the paragraph "There are several application using nanoparticles as a component to deliver physical-directed therapies. Several research groups have used 46 nanoparticles for photothermal therapy using direct tumor injection and generating cell 47 ablation by excessive bulk heating" authors must discuss the limitation of intratumoral administration.
Previously to the description of Carbon nanoparticles (CNPs), the reviewer recommends to add a brief introduction about the different types of NPs.
Authors describe that CNPs "generate high surface temperatures (400°C)". Nevertheless, necrosis death pathway could lead tostimulate an inflammatory response and thus, promote tumor progression mediated, for example, by the Tumor Necrosis Factor-α (TNF-α) [10.1146/annurev.pathmechdis.3.121806.151456 , 10.1007/s10555-006-9005-3 , 10.1111/j.1745-7254.2008.00889.x] Please, discuss it.
The sentence "Not only such preparations are reversible, but the active mole- 75 cules are also not chemically modified so that they can act on the target in their native 76 forms." is true, but the reviewer recommends to discuss also that this type of interaction could promote the release of the drugs far away from the target site.
The paragraph "By exploiting the enhanced permeability and retention (EPR) effects via the tumor 86 vasculature, researchers have used these methods to maximize the concentration and ther- 87 apeutic potential of nanoparticles by combining their innate properties with different 88 sources of stimulus" needs references. DOI: 10.2174/1381612821666150820100812 . Moreover, authors must discuss the great controversy that currently exist about EPR DOI: 10.1016/j.jconrel.2014.03.057
The reviewer has disagreements over figure 1. The lower pH that has been describe for tumor microenvironment (up to my knowledge) is more than 6, even for high aggressive Glioblastoma Multiforme. DOI: 10.1002/nbm.1701 Please add references for pH 5 or correct it.
In the paragraph "Depending on the nanomaterial, photothermal effects 138can be induced upon light excitation and thereby cause local thermal coagulation and tis- 139 sue necrosis without damaging the untargeted regions" necrosis is mentioned. As previously commented, this death pathway could lead to undesired effect. Please, take it into account.
In the paragraph "Unlike light-based thermal therapies, oscillating magnetic fields 169 are used to stimulate IONPs which then subsequently generate frictional heat." oscillating is less commonly used. Please change to alternating.
The sentence "However, the surface of CNPs can be functional- 202ized, allowing them to become soluble in aqueous solutions." needs references
In the paragraph "Confocal fluorescence microscopy confirmed that 284
upon reaching the target, the functionalized CNTs were internalized and that a 5-minute 285 exposure to an 808 nm laser with a power of 3.5 W/cm2 was sufficient to inflict irreversible 286 damage on the tumor." authors refers to cells or a real tumor tisse? please confirm, since they are completly different.
In table 1. It would be recommended a new column with the administration route (at least for in vivo experiment). Moreover, Ref 138. In time there is not unit (10 min, 10 hours)...
The sentence "Lastly, targeting moieties specific to the tumors may also 320 be engineered onto the surface of the nanoparticles, promoting further specificity in drug 321 delivery thus minimizing potential cytotoxic side effects on non-cancerous cells." needs further discussion. Tumor targeting is not same as cell targeting. In table 1, majority of targeting molecules are cell targeting, not tumor targeting. Please clarify it.
The paragraph "Unlike graphene and carbon nanotubes, the size distribution of fullerenes is 351 relatively uniform, and the sizes of the common fullerenes in biomedical applications are 352 often smaller than 5nm, allowing these nanoparticles to passively accumulate at the tumor 353 microenvironment" needs further comments. 5 nm in size is lower than the cut-off of renal filtration DOI: 10.1038/nbt1340
The reviewer recommends to clarify section 4. The title of the work is "Functionalized Carbon Nanoparticles as Theranostic Agents 2 and their Future Clinical Utility in Oncology". Section 4 is "4. Potential Translation Of Nanoparticle Theranostics Into The Clinic". In section 4 there is only 2 CNPs and they are non functionalized... Therefore is out of the object of the review.
Author Response
The paragraph "However, in recent years, the utilization of nanoparticles has extended beyond the 40classical drug delivery methods to exploit their innate properties. Such unique properties 41of nanomaterials include but are not limited to, utilizing their high surface area to volume 42ratio for enhanced cargo loading, enhanced magnetic properties, and unique optical char- 43acteristics, which may be further augmented by external stimuli such as pH, temperature, 44light, and magnetic fields." needs references. The reviewer recommends: DOI:10.1021/acsami.1c02338 and DOI: 10.3390/pharmaceutics13030416
The references have been added.
In the paragraph "There are several application using nanoparticles as a component to deliver physical-directed therapies. Several research groups have used 46 nanoparticles for photothermal therapy using direct tumor injection and generating cell 47 ablation by excessive bulk heating" authors must discuss the limitation of intratumoral administration.
Direct intratumoral injection does have its technical limitations. Consequently, as a result of intravenous injection and targeted accumulation, near-field photoablation phenomena is achieved because an effective concentration of nanoparticles reaches the tumor site. The damage that photothermal therapy aims to elicit, is not a result of excessive bulk heating but near-field (i.e., near the surface of the targeted cell). This near-field effect is a result, as heat generated and dissipated is linked to the diameter of nanoparticle. However, intratumoral injection would result in an “extra” fold-greater concentration of nanoparticles. If same laser power and durations are used, it would result in excessive bulk heating. Bulk heating would result in tumor ablation but would also lead to extended and wanted damage or normal tissue and result in serious injury. Therefore, we have revised our manuscript to include this discussion.
Previously to the description of Carbon nanoparticles (CNPs), the reviewer recommends to add a brief introduction about the different types of NPs.
This is something we considered when initially drafting our manuscript. Including an introduction into other types of nanoparticles, even if limited to inorganic-based nanoparticles, would have lengthened our manuscript. A reviewer has already commented on shortening our introduction.
Our aim was to focus our discussion on different CNP types and their therapeutic and imaging utilities. Therefore, within the manuscript, we have included descriptions of the application of the theranostic properties of gold- and iron-based nanoparticles. This also includes extensive description of clinical trials, an area where CNP is currently lacking.
Authors describe that CNPs "generate high surface temperatures (400°C)". Nevertheless, necrosis death pathway could lead to stimulate an inflammatory response and thus, promote tumor progression mediated, for example, by the Tumor Necrosis Factor-α (TNF-α) [10.1146/annurev.pathmechdis.3.121806.151456,10.1007/s10555-006-9005-3 , 10.1111/j.1745-7254.2008.00889.x] Please, discuss it.
The effects of pro-inflammatory responses elicited by photo-therapies are currently under investigation by various groups. While some of these inflammatory responses may cause undesired effects, they have been also utilized to promote T-cell mediated anti-tumor responses to further augment cancer treatment in other cases. We believe such discussions on inflammatory responses are beyond the scope of this review, however, we have included a brief description of the immune pathway activation from nanoparticle-induced phototherapy.
The sentence "Not only such preparations are reversible, but the active mole- 75 cules are also not chemically modified so that they can act on the target in their native 76 forms." is true, but the reviewer recommends to discuss also that this type of interaction could promote the release of the drugs far away from the target site.
The sentence has been modified to highlight the possibility that the non-covalent, reversible loading of cargo molecules can promote their release away from the target site, reducing the concentration of the payload reaching the tumor site, but also potentially causing unwanted side effects.
The paragraph "By exploiting the enhanced permeability and retention (EPR) effects via the tumor 86 vasculature, researchers have used these methods to maximize the concentration and ther- 87 apeutic potential of nanoparticles by combining their innate properties with different 88 sources of stimulus" needs references. DOI: 10.2174/1381612821666150820100812 . Moreover, authors must discuss the great controversy that currently exist about EPR DOI: 10.1016/j.jconrel.2014.03.057
We acknowledge that there have been clinical observations of the limitations of EPR, and the suggested review article highlights the complications associated with the EPR model. It’s clear it comes down to a case-for-case and the effectiveness of EPR. However, rather than discussing the controversy of this topic, we have revised the paragraph to place an emphasis on two important factors that are required for the nanoparticle to be an effective theranostic platform. These factors include tumor accumulation and retention. We go on to discuss properties required by the nanoparticle and tumor physiology. As such, EPR is one of the tumor properties able to enhance nanoparticle accumulation and retention.
The reviewer has disagreements over figure 1. The lower pH that has been describe for tumor microenvironment (up to my knowledge) is more than 6, even for high aggressive Glioblastoma Multiforme. DOI: 10.1002/nbm.1701 Please add references for pH 5 or correct it.
We have gone ahead and corrected the figure, to illustrate a pH range of 5.5 – 6.5, including the reference for Glioblastoma, that has experimental measured a mean pH range for Glioblastoma tumors of 6.5. We have included other review references, that indicate lower pH ranges for tumor microenvironments (10.1146/annurev-physiol-021119-034627, 10.1021/acsami.8b22487, 10.3389/fphys.2013.00354).
In the paragraph "Depending on the nanomaterial, photothermal effects 138can be induced upon light excitation and thereby cause local thermal coagulation and tis- 139 sue necrosis without damaging the untargeted regions" necrosis is mentioned. As previously commented, this death pathway could lead to undesired effect. Please, take it into account.
In the paragraph "Unlike light-based thermal therapies, oscillating magnetic fields 169 are used to stimulate IONPs which then subsequently generate frictional heat." oscillating is less commonly used. Please change to alternating.
Revision have been made.
The sentence "However, the surface of CNPs can be functional- 202ized, allowing them to become soluble in aqueous solutions." needs references
We have added the references accordingly (10.1039/D1NA00293G, 10.3390/c5040072)
In the paragraph "Confocal fluorescence microscopy confirmed that 284 upon reaching the target, the functionalized CNTs were internalized and that a 5-minute 285 exposure to an 808 nm laser with a power of 3.5 W/cm2 was sufficient to inflict irreversible 286 damage on the tumor." authors refers to cells or a real tumor tisse? please confirm, since they are completly different.
The reference is specific to HeLa cells and has been revised.
In table 1. It would be recommended a new column with the administration route (at least for in vivo experiment). Moreover, Ref 138. In time there is not unit (10 min, 10 hours)...
The unit of time has been added.
We have also included a column identifying whether this is an in vitro vs. in vivo experiment. For in vivo experiments we have defined the route of administration (e.g., intravenous vs. intratumoral).
The sentence "Lastly, targeting moieties specific to the tumors may also 320 be engineered onto the surface of the nanoparticles, promoting further specificity in drug 321 delivery thus minimizing potential cytotoxic side effects on non-cancerous cells." needs further discussion. Tumor targeting is not same as cell targeting. In table 1, majority of targeting molecules are cell targeting, not tumor targeting. Please clarify it.
We have provided the needed clarity by differentiating between in vivo vs. in vitro cell-based experiments.
The paragraph "Unlike graphene and carbon nanotubes, the size distribution of fullerenes is 351 relatively uniform, and the sizes of the common fullerenes in biomedical applications are 352 often smaller than 5nm, allowing these nanoparticles to passively accumulate at the tumor 353 microenvironment" needs further comments. 5 nm in size is lower than the cut-off of renal filtration DOI: 10.1038/nbt1340
We have included a statement, with reference, that because of the smaller size of fullerenes vs. CNTs/graphene they would also be rapidly cleared from the body by renal filtration and urinary excretion and minimize nanoparticle toxicity.
The reviewer recommends to clarify section 4. The title of the work is "Functionalized Carbon Nanoparticles as Theranostic Agents 2 and their Future Clinical Utility in Oncology". Section 4 is "4. Potential Translation Of Nanoparticle Theranostics Into The Clinic". In section 4 there is only 2 CNPs and they are non functionalized... Therefore is out of the object of the review.
The aim of section 4 was to provide a discussion of the clinical utility of inorganic- and synthetic-based nanoparticles for targeted tumor therapy. We have revised the manuscript to include a statement referring to the limited clinical trials using CNPs. The majority of clinical trials appear to be using carbon nanoparticles as a contrast tracer during surgeries. There is not adequate information provided on the CNP type, however, we did identify and referred 4 clinical trials using CNTs and quantum dots in either imaging or therapeutic applications.
As a complement, we have provided a list of advantages CNTs would have in the clinical arena. Furthermore, we have provided significant discussion of the pre-clinical application of CNPs in section 3, followed by section 4, which is a clinical translation of that discussion. Therefore, we don’t believe it is out of objective with the review. We have gone ahead to ensure that our CNP references are formulated for tumor targeting.
Round 2
Reviewer 2 Report
I agree with the publication of this review article.